# Variations in antibiotic prescribing among village doctors in a rural region of Shandong province, China: a cross-sectional analysis of prescriptions

Oliver James Dyar ,[1] Ding Yang,[2,3] Jia Yin,[2,3] Qiang Sun,[2,3] Cecilia Stålsby Lundborg[1]

OJD and DY contributed equally.

[1]Department of Global Public Health, Karolinska Institutet, Stockholm, Sweden
[2]School of Health Care Management, Cheeloo College of Medicine, Shandong University, Jinan, Shandong, China
[3]NHC Key Laboratory of Health Economics and Policy Research, Shandong University, Jinan, Shandong, China

**Correspondence to**
Professor Qiang Sun;
qiangs@sdu.edu.cn

## ABSTRACT

**Objectives** To assess variation in antibiotic prescribing practices among village doctors in a rural region of Shandong province, China.

**Design, setting and participants** Almost all outpatient encounters at village clinics result in a prescription being issued. Prescriptions were collected over a 2.5-year period from 8 primary care village clinics staffed by 24 doctors located around a town in rural Shandong province. A target of 60 prescriptions per clinic per month was sampled from an average total of around 300. Prescriptions were analysed at both aggregate and individual-prescriber levels, with a focus on diagnoses of likely viral acute upper respiratory tract infections (AURIs), defined as International Classification of Diseases, 10th Revision codes J00 and J06.9.

**Main outcome measures** Proportions of prescriptions for AURIs containing (1) at least one antibiotic, (2) multiple antibiotics, (3) at least one parenteral antibiotic; classes and agents of antibiotics prescribed.

**Results** In total, 14 471 prescriptions from 23 prescribers were ultimately included, of which 5833 (40.3%) contained at least 1 antibiotic. Nearly two-thirds 62.5% (3237/5177) of likely viral AURI prescriptions contained an antibiotic, accounting for 55.5% (3237/5833) of all antibiotic-containing prescriptions. For AURIs, there was wide variation at the individual level in antibiotic prescribing rates (33.1%–88.0%), as well multiple antibiotic prescribing rates (1.3%–60.2%) and parenteral antibiotic prescribing rates (3.2%–62.1%). Each village doctor prescribed between 11 and 21 unique agents for AURIs, including many broad-spectrum antibiotics. Doctors in the highest quartile for antibiotic prescribing rates for AURI also had higher antibiotic prescribing rates than doctors in the lowest quartile for potentially bacterial upper respiratory tract infections (pharyngitis, tonsillitis, laryngopharyngitis; 89.1% vs 72.4%, p=0.002).

**Conclusions** All village doctors overused antibiotics for respiratory tract infections. Variations in individual prescriber practices are significant even in a small homogenous setting and should be accounted for when developing targets and interventions to improve antibiotic use.

## BACKGROUND

Irrational antibiotic use is an important and modifiable driver of antibiotic resistance.[1]

### Strengths and limitations of this study

► We investigated whether significant variations exist in antibiotic prescribing patterns among village doctors working in a small, homogenous setting in rural China.
► We included over 14 000 prescriptions sampled across a 2.5-year time period.
► Uniquely, we analysed prescriptions both according to clinical diagnosis, focussing on acute upper respiratory tract infections, and at the level of individual prescribers.
► We could not verify diagnoses clinically, so variations between the village doctors in terms of diagnostic ability could not be assessed.

A knowledge of the specific patterns of unnecessary or inappropriate antibiotic use is important for designing context-adapted, targeted interventions to improve antibiotic use, such as education focussing on common irrational practices within a region or institution.[2] Understanding the variations that occur in antibiotic prescribing patterns at the prescriber level can further be valuable for shedding light on the facilitators and barriers to rational antibiotic use, as well as for enabling individualised feedback on prescribing practices.[3 4]

Previous studies in China strongly indicate that there is widespread antibiotic overuse in outpatient clinics in rural areas, particularly for respiratory tract infections (RTIs).[5–8] These studies have been less capable of investigating the extent of variability in practices that exist due to two common limitations: first, most studies have relied on highly aggregated data (eg, at the level of an institution or county); second, some studies have not been able to couple antibiotic prescriptions with clinical diagnoses at the level of the individual prescription. Recent studies in other

countries have highlighted the value of such finer-grained analyses; for example, Pouwels *et al* found that variability in prescribing patterns for antibiotics at primary care centres in the UK could not be adequately explained by variations in patient consultation rate or pre-existing comorbidities,[9] that is, individual prescriber behaviours are an important cause of irrational antibiotic use, and so need to be addressed.[10]

We investigated patterns of antibiotic prescriptions at the individual prescriber level in eight village clinics in rural Shandong province, China, over a two-and-a-half-year time period. Village clinics are the first place that most rural residents seek healthcare, and almost all consultations result in patients receiving a prescription, in part because prescriptions are a way in which reimbursements for healthcare expenditures are calculated and administered. The clinics are staffed by village doctors who typically have a high school level education and additional vocation-related training; many of the older village doctors were trained under the barefoot doctor programme.[11] Our aims were to assess if there was overall evidence of irrational antibiotic use, and whether there was significant variability between the practices of individual prescribers in this small, homogenous region. Our study forms part of a large cross-sectoral research programme on antibiotic consumption and resistance involving Chinese and Swedish researchers, the Sino-Swedish Integrated Multisectoral Partnership for Antibiotic Resistance Containment (IMPACT).[12]

## MATERIALS AND METHODS
### Study design and setting
We conducted a prospective observational analysis of outpatient prescriptions from January 2015 to July 2017 at eight village clinics clustered around a single town in Z County, Shandong province. This county is in the middle level in Shandong province in terms of economics, health indicators and population size. As previously described in the full study protocol,[12] the county and town were first selected based on the requirements and design of the IMPACT research programme, in particular being broadly representative of rural Shandong province, and including the presence of local collaborators who were capable of providing administrative support throughout the entire duration of the research programme. Twelve villages were then purposively selected around the town from among 17 that had at least 100 households. These 12 villages are served by 8 village clinics located in the villages and are attended by 24 doctors who do not move between clinics.

### Data collection
The prescriptions of all village clinics in Z county are included in an e-prescription system that records for each visit the patients' name, sex, age, date of visit, diagnosis, medicines prescribed, total cost for the visit and name of the doctor. For each village clinic, a target of 60 prescriptions per month (from an average total of around 300) was collected using a random sampling method in which every fifth consecutive prescription was selected, beginning with a randomly generated starting number between 1 and 10. The prescription details were exported one by one from the e-prescription system in XPS format and entered into a Microsoft Excel database. In addition, in July 2015, the doctors completed a short face-to-face questionnaire concerning their demographic information (gender, age, number of years working).

### Data management and analyses
Prescriptions from one doctor who did not complete the questionnaire were removed (55 in total). Data from prescriptions from the remaining 23 doctors were then anonymised with patient names being removed and each doctor assigned a three-digit code representing the village clinic at which they worked (1XX–8XX) and a unique number (X01–X23).

All diagnoses on prescriptions were coded where possible according to the International Classification of Diseases, 10th Revision (ICD-10) by an independent researcher from China with a background in clinical medicine and public health: first, a list of unique diagnoses included on the prescriptions was created; second, these were matched with the standard ICD-10 Chinese translation provided by the National Health Commission of China that is used in hospitals; finally, these diagnoses were translated into English. In cases of uncertainty, a shortlist of potential diagnoses was discussed between the first author (a clinical doctor and public health researcher from Europe) and the independent researcher until either agreement was reached or no ICD-10 code was assigned (eg, when a symptom was listed instead of a diagnosis, or when it was unclear whether a diagnosis was acute or chronic); in total, these cases of uncertainty accounted for <10% of all prescriptions, and did not include any prescriptions that were ultimately classified as likely viral acute upper respiratory tract infections (AURIs). A category of *likely viral* AURIs was created by grouping diagnoses of J00 (acute nasopharyngitis (common cold) and J06.9 (acute upper respiratory infection, unspecified).

Prescriber-level analyses were limited to the 20/23 doctors with ≥50 AURI prescriptions during the study period, to ensure that the antibiotic prescribing rates (APRs) calculated for each individual doctor were sufficiently representative of their practice. Of the three doctors with fewer than 50 AURI prescriptions, two retired from clinical practice near the beginning of the study period (616 and 720), and one newly started working at the specific village clinic at the end of the study period (618). We further identified an error in the coding of prescriptions in village clinic 1 which was due to one doctor (101) being responsible for entering the majority of prescriptions onto the electronic system using their account, irrespective of who the actual prescriber was, because of the low computer literacy of the other doctor (102). We therefore could not be confident of

which doctor was responsible for which prescription, so all 433 prescriptions from the two doctors working at this clinic were excluded from prescriber-level analyses. This error was not present at other village clinics. All prescribed antibiotics were coded using the Anatomical Therapeutic Chemical (ATC) classification system[13] and categorised by classes and substances. Prescribed medicines were also coded as: 'Analgesic or anti-inflammatory medicines' (aspirin/acetylsalicylic acid, ibuprofen, paracetamol or diclofenac); 'traditional Chinese medicines'; or 'other'.

For each prescriber, antibiotic prescribing was analysed using three indicators: (1) Antibiotic prescribing rate (APR)=number of prescriptions that included at least one antibiotic/total number of prescriptions × 100%, (2) multiple antibiotics prescribing rate (MPR)=number of prescriptions that included at least two antibiotics with different ATC codes/total number of prescriptions that included at least one antibiotic × 100%, (3) parenteral APR (PAPR)=number of prescriptions that included at least one parenteral antibiotic/total number of prescriptions that included at least one antibiotic × 100%. Comparisons were also made against antibiotic prescribing quality indicators developed in Europe[14] for a subset of clinical diagnoses and patient ages.

All analyses were conducted in Microsoft Excel and SPSS (version 24). Descriptive statistics were calculated and comparisons were made using $\chi^2$ tests and Pearson correlation. The Kruskal-Wallis H-test was used to assess whether there were statistically significant variations between the individual doctors in terms of APRs for *likely viral AURIs*. Pairwise Mann-Whitney U-tests were used post-hoc to assess for differences in all pairwise comparisons between doctors, using the Benjamini-Hochberg procedure[15] to control the false discovery rate for multiple comparisons. Statistical significance was set at p<0.05 for all tests. We used the Strengthening the Reporting of Observational Studies in Epidemiology guidelines for reporting the results of cross-sectional observational study (see online supplementary file).

### Patient and public involvement

Patients or the public were not involved in the design, conduct, reporting or dissemination plans of this specific study, but have been involved in the dissemination plans for results of the overall IMPACT research programme.

## RESULTS
### Overview of prescriptions and diagnoses

In total, 14 471 prescriptions from 23 prescribers from January 2015 to July 2017 were included in the analyses. This included an average of 58 prescriptions per month per village clinic (mean range from 56 to 60). There was a total of 222 diagnoses in Chinese, with 20 diagnoses accounted for nearly 90% of all prescriptions (12 811/14 471). The 5 most common diagnoses overall were: J00 (acute nasopharyngitis (common cold),

34.1%, 4938/14 471), I10 (essential hypertension, 13.6%, 1966/14 471), K29.7 (gastritis, unspecified, 5.9%, 856/14 471), M13.9 (arthritis, unspecified, 5.9%, 955/14 471) and I25.1 (atherosclerotic heart disease, 3.2%, 470/14 471). Overall, 40.3% of the prescriptions (5833/14 471) contained at least 1 antibiotic.

Table 1 summarises the infection-related diagnoses that had a minimum of 10 prescriptions during the study period. Overall, RTIs accounted for 68.4% (3990/5833) and gastrointestinal conditions for 14.0% (815/5833) of all prescriptions containing at least 1 antibiotic. A total of 5177 prescriptions were categorised as *likely viral AURIs*, and 62.5% (3237/5177) of these prescriptions contained at least 1 antibiotic, accounting for 55.5% (3237/5833) of all antibiotic-containing prescriptions. Fifteen per cent (491/3237) of AURI antibiotic prescriptions included two or more antibiotics and 24.3% (785/3237) included at least one parenteral antibiotic.

### Medicines prescribed for likely viral AURIs

Figure 1 shows the combinations of different medicine types (antibiotics, anti-inflammatories or analgesics, and traditional Chinese medicines) prescribed for AURI. Approximately half of all AURI prescriptions (54.0%, 2795/5177) contained only one medicine type, a third (33.7%, 1744/5177) contained two medicine types and 8% (414/5177) contained all three medicine types.

Sixteen classes of antibiotics and 27 antibiotic agents were prescribed for AURI (online supplementary appendix table 1). The three most common classes of antibiotics prescribed were: J01CA (penicillins with extended spectrum, 31.6% of prescriptions), J01DB (first-generation cephalosporins, 18.6% of prescriptions), J01FA (macrolides, 15.7% of prescriptions). The three most common agents of antibiotics prescribed for AURI were J01CA04 (amoxicillin, 31.4% of prescriptions), J01MA12 (levofloxacin, 10.3% of prescriptions) and J01DB01 (cefalexin, 9.9% of prescriptions).

### Comparison with European quality indicators

Table 2 shows APRs and types of antibiotic prescribed for a variety of RTIs in our dataset, together with previously published European recommendations.[14]

### Prescriber demographics

Table 3 summarises prescribers' demographic information and APRs for patients with AURI. The median age of the prescribers was 51 years old, and the median number of years of working experience was 31. Male prescribers were more likely to prescribe antibiotics for their patients than female prescribers (67.7% vs 42.7%, p<0.01). Male prescribers were slightly more likely to prescribe antibiotics for female patients than for male patients (69.5% vs 66.3%, p=0.04), but no difference was seen among female prescribers (41.9% vs 43.4%, p=0.65).

### Variations at the prescriber level

Overall, individual prescribers' AURI APRs were highly correlated between 2015 and 2016 (r=0.65). The mean

**Table 1** Infection-related diagnoses and antibiotic prescription rates

| Body system (no. of prescriptions) | Diagnosis | ICD-10 code | Antibiotic prescribing rate | |
|---|---|---|---|---|
| | | | n/N | % |
| Respiratory (6178) | Acute nasopharyngitis (common cold)† | J00 | 3034/4938 | 61.4 |
| | Bronchitis* | J40 | 408/522 | 78.2 |
| | Acute upper respiratory infection† | J06.9 | 203/239 | 84.9 |
| | Acute pharyngitis | J02.9 | 160/190 | 84.2 |
| | Acute tonsillitis | J03.9 | 68/73 | 93.2 |
| | Rhinitis* | NA | 31/67 | 46.3 |
| | Cough | NA | 23/65 | 35.4 |
| | Pneumonia | J18 | 40/51 | 78.4 |
| | Bronchopneumonia | J18.0 | 15/19 | 78.9 |
| | Acute laryngopharyngitis | J06.0 | 8/14 | 57.1 |
| Gastrointestinal (1642) | Gastritis | K29.7 | 258/856 | 30.1 |
| | Gastroenteritis | A09.9 | 496/717 | 69.2 |
| | Diarrhoea | NA | 61/69 | 88.4 |
| Dental (222) | Chronic periodontitis | K05.3 | 100/131 | 76.3 |
| | Gingivitis and periodontal diseases | K05 | 61/72 | 84.7 |
| | Pulpitis | K04.0 | 19/19 | 100.0 |
| Urogenital (84) | Inflammatory disease of prostate | N41.9 | 23/45 | 51.1 |
| | Urethritis and urethral syndrome | N34 | 23/24 | 95.8 |
| | Urinary tract infection | N39.0 | 15/15 | 100.0 |
| Eye (101) | Conjunctivitis | H10 | 9/101 | 8.9 |
| Ear (11) | Otitis media | H66.9 | 7/11 | 63.6 |

*Not specified as acute or chronic.
†Categorised as likely viral acute upper respiratory tract infections.
APR, antibiotic prescribing rate; ICD-10, International Classification of Diseases, 10th Revision; NA, no ICD-10 code was allocated.

absolute individual variation in AURI APRs between 2015 and 2016 was ±13.1% (SD=6.7%). Eleven prescribers had lower AURI APRs in 2016 compared with 2015, and six had higher APRs in 2016 (online supplementary appendix table 2).

Amoxicillin was the most commonly prescribed agent for AURI for 13/18 doctors; the other agents that were the single most commonly prescribed for individual doctors included levofloxacin (2 doctors), amikacin (1), lincomycin (1) and azithromycin (1). For each doctor, their three most commonly prescribed antibiotic agents accounted for a mean of 64.4% (range 45.3%–86.4%) of all the antibiotics they prescribed for AURI (online supplementary appendix table 3).

Figure 2 and online supplementary appendix table 4 illustrate the individual variations between the 18 doctors for AURI prescriptions. There was wide variation in the APR (33.1%–88.0%), MPR (1.3%–60.2%) and PAPR (3.2%–62.1%). Village doctors' APRs were positively correlated with their MPRs ($r$=0.47, p=0.05) and PAPRs ($r$=0.54, p=0.02), but there was no association between doctors' APRs and the prescription of at least one analgesic

or anti-inflammatory medicine ($r$=−0.19, p=0.46). Village doctors' MPRs were also positively correlated with their PAPRs ($r$=0.77, p<0.01).

The Kruskal-Wallis H-test showed that the variations in APRs for AURI prescriptions were statistically significant among the 18 prescribers (p<0.001). Pairwise Mann-Whitney U-tests were used as post-hoc tests to assess for differences in all possible pairwise comparisons between prescribers. This involved 153 pairs of prescribers, and the difference was statistically significant in 93 of these.

Figure 3 and online supplementary appendix table 5 illustrate the different types of medicines used for AURI prescriptions by 18 prescribers. The most common choice for prescribers (14/18) was to use only an antibiotic; the most common choice for the four remaining prescribers was to use only analgesic or anti-inflammatory medicines. All prescribers prescribed all three types of medicines for patients with a diagnosis of AURI on at least one occasion.

**Comparisons between high and low antibiotic prescribers**

Table 4 shows comparisons for selected indicators between the four prescribers with the highest APR for AURI (513,

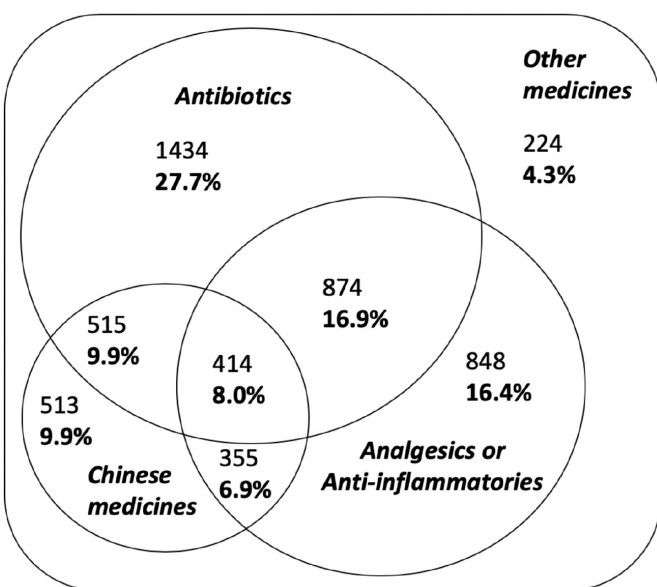

**Figure 1** Medicine types prescribed for likely viral acute upper respiratory tract infections. Analgesics or anti-inflammatories=any of the following: aspirin/acetylsalicylic acid, ibuprofen, paracetamol or diclofenac.

412, 204 and 206) and the four prescribers with the lowest APR for AURI (203, 823, 308, 307).

## DISCUSSION

We prospectively investigated patterns of antibiotic prescriptions over a two-and-a-half-year period at village clinics in rural Shandong province in order to assess evidence of irrational antibiotic use and variability in prescribing practices at the level of the individual village doctor.

### Consultation patterns and overall antibiotic use

The patterns of diagnoses recorded on the prescriptions in our study are broadly similar to those in previous studies conducted in rural China,[5 16] with RTIs accounting for a large proportion of patient visits. We found that

urogenital infections and skin and soft-tissue infections each accounted for less than 1% of all antibiotic prescriptions at the village clinics, whereas these conditions are typically the second and third most common reasons for antibiotic prescriptions in studies in primary care clinics from high-income countries.[17] In contrast, dental conditions were the third largest group of diagnoses for which antibiotics were prescribed in our study. This may reflect the extreme paucity of dentists in rural China,[11] and is potentially an important and under-reported source of antibiotic overuse. Overall, 40% of prescriptions contained at least one antibiotic. This antibiotic prescription rate is considerably higher than the government's target from 2012 of fewer than 20% outpatient prescriptions (for secondary-level hospitals),[18 19] but is lower than previously published studies in China which have tended to be around 50%.[5 7 16 20 21]

### Antibiotic use for RTIs

Although the antibiotic prescription rate is high in absolute terms, supporting clinical information is required to better understand the quality of the prescribing. We found that two-thirds of all antibiotic use occurred in patients with a diagnosis of an RTI, and that likely viral AURIs accounted for over 80% of the total antibiotic use for RTIs. This is despite there being very little evidence of benefit of antibiotic use for such AURIs, and a considerable number of strong recommendations that they should be avoided.[22 23] The APR for AURIs of 60% is similar to previous studies in rural village clinics in Shandong province,[8] Anhui province[24] and in Western China[16] as well as reports from rural India,[4 25] and Malaysia,[26] but far higher than rates observed in national-level studies in Sweden (8%), Belgium (19%) and Netherlands (38%).[27]

Antibiotic prescribing quality indicators were recently developed in Europe to assess both the decision to prescribe an antibiotic in specific clinical conditions and the actual choice of antibiotic agent.[14] All of our results lay outside of the recommended ranges. In particular, we found that the recommended J01CE antibiotics were

**Table 2** Antibiotic prescribing quality indicators for selected diagnoses

| Diagnosis | Patient age group (years) | APR n/N, % | Class of antibiotics | | |
| | | | J01M n/N, % | J01CE n/N, % | J01AA or J01CA n/N, % |
|---|---|---|---|---|---|
| AURI | ≥2 | 3236/5172, 62.6% | 436/3236, 13.5% | 49/3236, 1.5% | |
| *Target* | | *0%–20%* | *0%–5%* | *80%–100%* | *NA* |
| Tonsillitis | ≥2 | 68/73, 93.2% | 15/68, 22.1% | 2/68, 2.9% | |
| *Target* | | *0%–20%* | *0%–5%* | *80%–100%* | *NA* |
| Pneumonia | 18–65 | 22/30, 73.3% | 11/22, 50.0% | | 2/22, 12.8% |
| *Target* | | *90%–100%* | *0%–5%* | *NA* | *80%–100%* |

Targets are as previously published,[14] except for 'NA' which indicates that no target was included; J01M is the Anatomical Therapeutic Chemical (ATC) code for quinolone antibacterials, J01CE is the ATC code for beta-lactamase-sensitive penicillins, J01AA is the ATC code for tetracyclines, J01CA is the ATC code for penicillins with extended spectrum.
APR, antibiotic prescribing rate.

**Table 3** Prescriber demographics and antibiotic prescribing rates (APRs) for acute upper respiratory tract infections (AURIs)

| Prescriber | N | % | AURI prescriptions | | | |
|---|---|---|---|---|---|---|
| | | | N | APR (%) | MPR (%) | PAPR (%) |
| Total | | | 4710 | 63.0 | 15.7 | 26.0 |
| Gender | | | | | | |
| Male | 15 | 83.3 | 3821 | 67.7 | 17.1 | 28.4 |
| Female | 3 | 16.7 | 889 | 42.7 | 5.8 | 9.7 |
| Age | | | | | | |
| 36–39 | 2 | 11.1 | 823 | 65.7 | 18.7 | 26.1 |
| 40–49 | 6 | 33.3 | 1419 | 57.2 | 24.5 | 30.6 |
| 50–59 | 4 | 22.2 | 1114 | 63.6 | 8.2 | 27.4 |
| 60–68 | 6 | 33.3 | 1354 | 67.0 | 11.8 | 20.7 |
| Education* | | | | | | |
| Junior high school | 3 | 16.7 | 962 | 60.9 | 9.7 | 34.0 |
| Senior high school | 15 | 83.3 | 3748 | 63.5 | 17.1 | 24.0 |
| Working years | | | | | | |
| 14–19 | 5 | 27.8 | 1666 | 56.7 | 17.5 | 26.8 |
| 20–29 | 4 | 22.2 | 687 | 70.6 | 28.0 | 29.1 |
| 30–39 | 2 | 11.1 | 836 | 64.4 | 10.0 | 34.6 |
| 40–48 | 7 | 38.9 | 1521 | 65.7 | 11.0 | 19.1 |

*Education: junior high school only is equivalent to completing 9 years of full-time education; senior high school is equivalent to completing 12 years of full-time education.
APR, antibiotic prescribing rate; AURI, likely viral acute upper respiratory tract infections; MPR, multiple antibiotics prescribing rate; PAPR, parenteral antibiotics prescribing rate.

rarely used (1.5% for AURI, 2.9% for tonsillitis), whereas J01CA penicillins were used far more frequently. This may represent a prescriber preference for broader spectrum antibiotics.

Clinical treatment guidelines are not in widespread use in village clinics in China[28]; however, there are essential medicine lists and lists of antibiotics which require higher authorisation levels to be prescribed. All but one antibiotic prescribed for AURIs were either included on the Chinese national essential medicines list,[29] or on the list of supplemental essential medicines available in Shandong province; cefixime did not feature on either of these two lists and accounted for 1.6% of all antibiotics prescribed for AURIs during the study period. The high rate of parenteral antibiotics use, third-generation cephalosporins and fluroquinolones in our study are concerning, but

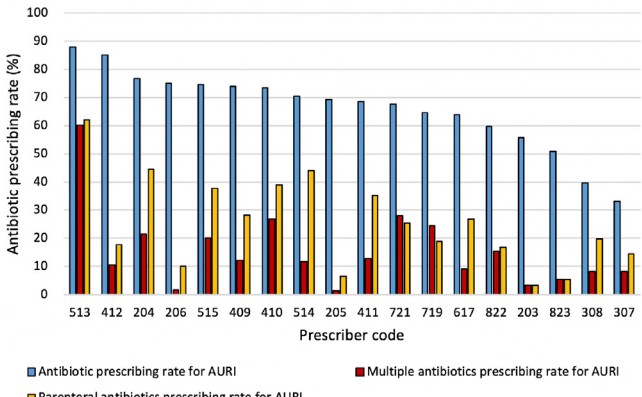

**Figure 2** Antibiotic prescribing rates for likely viral upper respiratory tract infections (AURIs) for individual prescribers

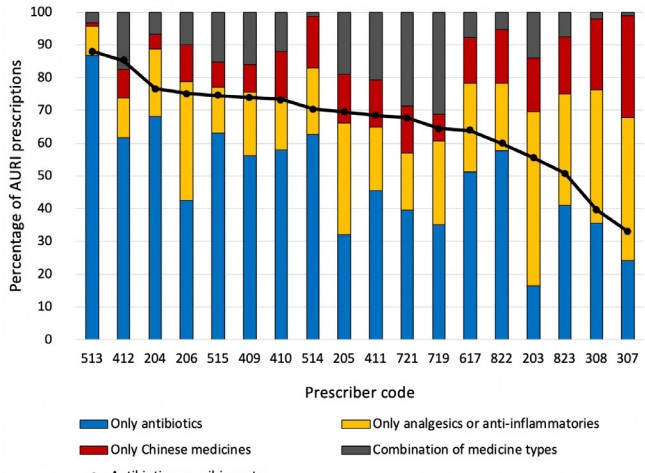

**Figure 3** Different medicine types used for likely viral acute upper respiratory tract infections (AURIs) by individual prescribers.

**Table 4** Variation between high and low acute upper respiratory tract infection (AURI) antibiotic prescribing rate (APR) groups

| Indicator | High AURI APR group (%) | Low AURI APR group (%) | $\chi^2$ value | P value |
|---|---|---|---|---|
| Mean APR for likely viral AURIs | 81.3 | 44.8 | 273.31 | <0.001 |
| Mean MPR for likely viral AURIs | 23.4 | 6.2 | 36.06 | <0.001 |
| Mean PAPR for likely viral AURIs | 33.6 | 10.7 | 44.57 | <0.001 |
| Mean prescribing rate of only antibiotics for likely viral AURIs | 40.6 | 18.3 | 72.88 | <0.001 |
| Mean prescribing rate of only analgesics or anti-inflammatory medicines for likely viral AURIs | 10.6 | 26.2 | 74.26 | <0.001 |
| Mean APR for URTI with potential bacterial causes*† | 89.1 | 72.4 | 9.25 | 0.002 |
| Mean APR for gastritis, gastroenteritis and diarrhoea† | 85.0 | 68.3 | 9.17 | 0.002 |

The high acute upper respiratory tract infection (AURI) APR group contained the four village doctors with the highest antibiotic prescribing rates for likely viral AURIs, and the low AURI APR group contained the four village doctors with the lowest antibiotic prescribing rates for likely viral AURIs.
*URTI with potential bacterial causes=prescriptions with an upper respiratory tract infection diagnosis of pharyngitis, tonsillitis or laryngopharyngitis.
†These comparisons were restricted to village doctors in each group who had at least 10 prescriptions containing a relevant diagnosis.
APR, antibiotic prescribing rate; MPR, multiple antibiotics prescribing rate; PAPR, parenteral antibiotics prescribing rate.

are consistent with the results of province-wide antibiotic consumption data that were recently reported in Shandong province using procurement records,[30] as well as previous studies at healthcare institutions in rural Shandong province.[8] In a survey, we previously conducted at the village clinics included in this study, 74% of doctors reported thinking that parenteral antibiotics are more effective than oral antibiotics.[31]

Interestingly, we found that antibiotics were used less frequently than recommended for adult patients with a diagnosis of pneumonia (73.3% compared with a target of 90%–100%),[14] potentially representing underuse of antibiotics. Analgesic and anti-inflammatory medicines are potentially underused, since they have the potential to both provide symptomatic relief and satisfy patients' expectations to gain some form of medicine from visiting the doctor; less than half of all patients with a diagnosis of AURI received such a medication.

### Variations in antibiotic prescribing between prescribers

Understanding what variations exist in clinical care, and why, is important for developing targets as well as context-specific interventions capable of reaching these targets. It is also essential for working towards individuals having equitable access to healthcare, regardless of where they reside.

Our study design has enabled us to investigate variability in antibiotic prescribing at the prescriber-level while keeping several factors relatively constant: the villages served by the clinics are limited to a small geographical region and are quite homogeneous in terms of socioeconomic levels; the patient populations in the villages are likely to have similar consultation behaviours when ill, in particular for a common diagnosis such as AURI; the doctors all have the same medical qualifications and at least 14 years of clinical experience; the village clinics are all subject to the same reimbursement systems (including zero markup on antibiotics) and regulations concerning antibiotic use.[32 33] Given these similarities, it might be hypothesised that little variation would exist in the prescribing practices between the doctors. Our results, however, revealed substantial differences between prescribers. Patients with AURI who were seen by the high-antibiotic prescribers were twice as likely to be given a prescription containing an antibiotic, four times as likely to be given two or more antibiotics and three times as likely to receive at least one parenteral antibiotic. There was also significant variation within the high-prescriber group, highlighting the importance of individual-level analyses: patients with AURI who visited doctors 513 and 412 were equally likely to be prescribed an antibiotic (mean APR of 87%), but those who visited doctor 513 were six times more likely to be prescribed two or more antibiotics, three times more likely to be prescribed an parenteral antibiotic and a third less likely to be prescribed an analgesic or anti-inflammatory medicine; furthermore, the choice of antibiotic prescribed was likely to be an aminoglycoside, a third generation cephalosporin or a fluoroquinolone, compared with amoxicillin, a first-generation cephalosporin, or a macrolide. Separately, a single doctor was responsible for a quarter of all amikacin use during the study period. Such extreme variation is important for individual patients in terms of risks of developing antibiotic-related complications and the impact of antibiotic use on their normal flora.[34] This ethical component should not be understated, and there is a growing trend for antibiotic stewardship to be viewed in terms of patient safety and broader responsibility.[35 36]

Relatively few published studies have assessed variations between prescribers concerning antibiotic use in primary care settings, with most studies analysing aggregated data and presenting averages. In 2001, Akkerman *et al* investigated APRs for upper respiratory tract infections (URTIs) among 84 primary care practitioners in the Netherlands, finding that these ranged from 15% to 27%.[37] More recently, Grover *et al* examined antibiotic prescribing for RTIs at a single outpatient practice in the USA and found variations depending on the provider type from 55% (residents) to 84% (nurse practitioners).[38] Jung *et al* found an IQR of 27%–60% in APRs for RTIs among 109 prescribers in a large healthcare network in the USA[39]; a study conducted in a separate US healthcare network of urgent care clinics found an IQR in APRs among physicians of 7%–28% for respiratory encounters where antibiotics are not indicated.[40] A tempting justification against conducting such individual-level analyses is that it is difficult to account for the variation in patient case mix. This does indeed limit the ability to make comparisons, particularly between studies conducted in settings in which patients have very different consultation rates and comorbidities, or in studies that look at overall antibiotic prescription rates rather than at a single diagnosis. However, there is growing evidence from these studies that differences in patient case mix may not explain a large amount of variability in prescribing patterns for antibiotics.[9 39] Furthermore, ignoring variations between individual prescribers causes us to subtly obscure the role (or even responsibility) of the individual decision-maker. Interestingly, the doctors in our study used between 11 and 21 different antibiotic agents for patients with AURI, demonstrating the heterogeneity even within an individual prescriber.

Future studies should explore why such variations exist, as well as what the consequences are for efforts to improve antibiotic use. For example, qualitative studies with individual prescribers could help understand why some doctors already have very low rates of parenteral and multiantibiotic use, and why prescribers have used such a wide range of antibiotic agents for patients with a single diagnosis. It is possible that structural factors such medicine shortages might contribute to changes in antibiotics prescribed. We did find that high antibiotic AURI prescribers were more likely than low antibiotic AURI prescribers to prescribe antibiotics for the group of URTI diagnoses that are most likely to have bacterial causes (but where the most common causes are still viral). One possible explanation is that these prescribers could be more risk averse, and so are more likely than other doctors to prescribe antibiotics in cases where the cause could be bacterial[41]; an alternative is that these prescribers are more willing than others to follow patient demand, particularly if this is generally high. Future studies could also explore the influence of peers within clinics; our results suggest that there is some clustering of behaviours within certain village clinics.

### Methodological considerations

Our study has several strengths, including the long length of data collection and the high number of prescriptions included by individual village doctors. Furthermore, almost all patients that visit doctors at a village clinic receive a prescription. A common limitation of studies using prescription data is the inability to verify diagnoses. To reduce the impact of this limitation, we restricted the individual-level analyses to AURI diagnoses, but we cannot guarantee that misclassifications did not occur, particularly given there is currently no incentive for doctors to correctly classify diagnoses. AURIs accounted on average for 86% of RTI diagnoses by individual doctors, but this ranged from 58% to 100% which strongly suggests that there were variations in how doctors classified different infections, and this warrants further investigation, particularly if diagnosis-based antibiotic prescribing targets are ever introduced. Indeed, a recent study concluded that deficits in diagnostic knowledge are a major driver of unnecessary antibiotic prescriptions in clinics in rural China.[42] Additional limitations include an inability to account for variations in the workload of individual doctors, and to control for patient comorbidities. Finally, only one diagnosis can be included per prescription on the electronic prescribing system. It is possible that some antibiotics prescribed for likely viral AURIs are actually treatment for other infections that patients presented with at the same time; however, doctors would usually choose to write the more severe diagnosis, so we believe that this is likely only relevant to a very small number of cases and will not have significantly affected our main findings.

### CONCLUSION

Together our results provide evidence that even in a small homogenous setting in rural China, variations in individual prescriber practices are significant, and they need to be accounted for in the development of targets and interventions to improve antibiotic use. We suspect that this finding is likely to be generalisable to other areas in rural Shandong province as a minimum, but also to other areas in eastern rural China, where the healthcare system is very similar in terms of structure, regulations and staffing. We show that there is a need for all prescribers to reduce their APR for AURIs, but the extent of this reduction varies widely between prescribers. For some prescribers, reducing the proportion of prescriptions that contain fluoroquinolones and third-generation cephalosporins may be more important goals, both for their patients and for public health. Finally, it is possible that interventions that incorporate individual-level feedback will have a more directed and effective impact on improving antibiotic use. A promising recent example in Anhui province, China, was able to provide 'just-in-time' individualised information and feedback to doctors

in village clinics which improved antibiotic prescribing practices.[43]

**Acknowledgements** The authors acknowledge the support of the local Centre for Disease Control and Prevention in coordinating this study, as well as all students who contributed to the data collection. They would also like to thank Yuanyuan Wang for assisting with the coding of diagnoses according to ICD-10 standards. Finally, they are grateful for the efforts of the IMPACT-consortium:https://www. folkhalsomyndigheten.se/impact/.

**Contributors** All authors were involved in the conception, design and implementation of the project. YD and JY participated in the data collection. OJD and YD analysed and interpreted the data, and drafted the manuscript, with JY, QS and CSL providing critical input at all stages. All authors approved the final version of the manuscript.

**Funding** This work was supported by the Swedish Research Council (grant number D0879801) and National Natural Science Foundation of China (grant number 81361138021).

**Competing interests** None declared.

**Patient and public involvement** Patients and/or the public were not involved in the design, or conduct, or reporting, or dissemination plans of this research.

**Patient consent for publication** Not required.

**Ethics approval** Informed consent was obtained from all village doctors. Ethical approval was obtained from the First Affiliated Hospital, College of Medicine, Zhejiang University, China (reference numbers 2015#185 and 2015#283).

**Provenance and peer review** Not commissioned; externally peer reviewed.

**Data availability statement** Data are available upon reasonable request. The data sets generated and analysed during the current study are available on reasonable request from the corresponding author.

**ORCID iD**
Oliver James Dyar http://orcid.org/0000-0002-0094-3303

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
