## [Reviewer comments · BMJ Open]

ARTICLE DETAILS

TITLE (PROVISIONAL)	Variations in antibiotic prescribing among village doctors in a rural region of Shandong province, China: a cross-sectional analysis of prescriptions
AUTHORS	Dyar, Oliver; Ding, Yang; Yin, Jia; Sun, Qiang; Lundborg, Cecilia

VERSION 1 - REVIEW

REVIEWER	Carl Llor University Institute in Primary Care Research Jordi Gol, Via Roma Health Centre, Barcelona. I report having received research grants from Abbott Diagnostics.
REVIEW RETURNED	21-Jan-2020

GENERAL COMMENTS	Extrapolation of the results: The title of the manuscript and its objective is 'Variations in antibiotic prescribing among village doctors in rural Shandong province, China: a cross-sectional analysis of prescriptions'. However, GPs were selected from twelve villages around a town in this province. How confident are you with the extrapolation of the results of this study to the whole province. You might have missed other socioeconomic realities in the same province, which could have led to different results. You should discuss this point more in depth. You also mention both in the study and in the protocol of the study that this selection was based on the requirements and design of the entire IMPACT research program. You should also explain this more thoroughly. The prescriber-level analysis was limited to doctors with more than 50 upper respiratory infection prescriptions. This could have biased the results obtained. Explain. There is no discussion about guidelines used by Chinese doctors. Some of the results found are not different from other countries and the wide variation across professionals is a characteristic observed in many other countries. However, the high rate of parenteral antibiotics, third-generation cephalosporins and fluroquinolones found in this study merits a discussion. You refer pharyngitis and laryngopharyngitis as possible bacterial infections (Table 4). Is that correct? You highlighted the high consumption of antibiotics for gastrointestinal and dental infections compared to studies carried out in high-income countries but no references are provided. Not many papers on dental infections have been published in western
--

	countries. Did you analyze doctors' characteristics when you compared the high and the low prescribers? For instance, use of rapid tests, time per consultation, etc.
--	---

REVIEWER	David Hyun Antibiotic Resistance Project, The Pew Charitable Trusts, United States
REVIEW RETURNED	22-Feb-2020

GENERAL COMMENTS	1. I recommend moving up the description included at the end of the manuscript in Methodological Considerations, that all clinic visits ends with a prescription, to the materials and methods section. As a reader not familiar with Chinese village clinics and their practices, I found myself questioning the authors' implied equivalence between the total number of prescriptions and the total number clinic visits for the denominator of the rates measured, until I found the clarification at the end of the manuscript. 2. A few clarifications will be helpful in the methods employed on translating the diagnoses captured from the e-prescription databases to ICD10 codes, given my unfamiliarity of how diagnoses are documented in the e-prescriptions from village clinics and since the crossmapping to ICD10 codes was conducted by just one researcher. a. What were the qualifications of the independent researcher who conducted this coding translation? I'm assuming it was someone with clinical training or background? b. Was there a pre-defined list/mapping of Chinese symptoms/diagnoses to ICD10 that the independent researcher used systematically? c. Does the e-prescription allow only one diagnoses entry from each visit? Other studies from other countries have shown that chart and billing documentation from outpatient visits, especially for acute respiratory conditions, frequently results in more than one diagnoses. And many of those multiple diagnoses have differing appropriateness for antibiotic prescribing (for instance URI and pneumonia diagnosis documented/coded in the same visit). Are there any limitations in the e-prescription diagnosis data that could suggest the possibility that certain clinic visits with multiple diagnoses may have included a diagnosis that was antibiotic appropriate that was not captured in the e-prescription system (such as if the e-prescription system auto-populates the first diagnosis from a list of diagnoses in the electronic medical chart)? d. Would like to know about the "coding error" that led to the exclusion of prescriber from the analysis – was it a coding error originated from the e-prescription system or was it a coding error by the independent researcher during the ICD10 code crossmapping process? 3. Reviewing the N values of the individual prescribers in Appendix Table 2, it would be helpful to have the following questions addressed in the text – mostly relating to additional information on the individual provider practice characteristics. a. Looking at the two excluded prescribers (616 and 618) due to their very low N values for AURIs, it raises the question on why their Ns were so low to begin with especially considering they were from
--

	the same clinic and the only other prescriber in the same clinic (617) was responsible for over 90% of the AURI visits. Were the two excluded prescribers part-time providers? Are they seasonal providers (doesn't see patients during the winter?) Are there any other demographics/practice factors that could account for this significance difference between providers in patient population within the same clinic? b. Are any of these factors applicable to other village clinics where it could have influenced patient population differences between the providers from the same clinic? 4. A couple of suggested bibliographies to add in the discussion around individual provider variation in antibiotic prescribing: Stenehjem E, Wallin A, Fleming-Dutra KE, Buckel WR, Stanfield V, Brunisholz KD, Sorensen J, Samore MH, Srivastava R, Hicks LA, Hersh AL. Antibiotic Prescribing Variability in a Large Urgent Care Network: A New Target for Outpatient Stewardship. Clin Infect Dis. 2019 Oct 23. pii: ciz910. doi: 10.1093/cid/ciz910. [Epub ahead of print] Jung S, Sexton ME, Owens S, Spell N, Fridkin S. Variability of Antibiotic Prescribing in a Large Healthcare Network Despite Adjusting for Patient-Mix: Reconsidering Targets for Improved Prescribing. Open Forum Infect Dis. 2019 Jan 18;6(2):ofz018. doi: 10.1093/ofid/ofz018.
--	---

VERSION 1 – AUTHOR RESPONSE

Reviewer(s)' Comments to Author:

Reviewer: 1

Reviewer Name: Carl Llor

Institution and Country: University Institute in Primary Care Research Jordi Gol, Via Roma Health Centre, Barcelona.

Please state any competing interests or state 'None declared': I report having received research grants from Abbott Diagnostics.

Comment: Extrapolation of the results: The title of the manuscript and its objective is 'Variations in antibiotic prescribing among village doctors in rural Shandong province, China: a cross-sectional analysis of prescriptions'. However, GPs were selected from twelve villages around a town in this province. How confident are you with the extrapolation of the results of this study to the whole province. You might have missed other socioeconomic realities in the same province, which could have led to different results. You should discuss this point more in depth. You also mention both in the study and in the protocol of the study that this selection was based on the requirements and design of the entire IMPACT research program. You should also explain this more thoroughly.

Response: Thank you, we agree this is an important point. We have revised our study objective and title so that it is clearer that our data collection is from a smaller rural region of Shandong province, rather than all of rural Shandong province: e.g. "Objective: To assess variation in antibiotic prescribing practices among village doctors in a rural region of Shandong province, China.". The introduction also now states that "We investigated patterns of antibiotic prescriptions at the individual prescriber-level in eight village clinics in rural Shandong province, China, over a two-and-a-half-year time period."

The study region was selected to be representative of rural Shandong province, as described in the study protocol. We write in the "study design and setting" section "This county is in the middle level in Shandong province in terms of economics, health indicators and population size. As previously

described in the full study protocol,[12] the county and town were first selected based on the requirements and design of the IMPACT research programme, in particular being broadly representative of rural Shandong province, and including the presence of local collaborators who were capable of providing administrative support throughout the entire duration of the research programme.”

Although there are indeed some socioeconomic variations in Shandong province, rural areas are generally quite homogenous, particularly in terms of population health, access to healthcare institutions, and the training of doctors working at healthcare institutions. This is seen in previous studies (such as Sun Q et al. BMC Pharmacol Toxicol. 2015;16:6). We believe that our main finding (the wide variation in antibiotic prescribing practices observed between individual village doctors) is likely to be generalisable to other parts of rural Shandong province, as a minimum, but also to other areas of eastern rural China, for which Shandong province is often chosen to be representative of. We have expanded in the conclusion section: “Together our results provide evidence that even in a small homogenous setting in rural China, variations in individual prescriber practices are significant, and they need to be accounted for in the development of targets and interventions to improve antibiotic use. We suspect this finding is likely to be generalizable to other areas in rural Shandong province as a minimum, but also to other areas in eastern rural China, where the healthcare system is very similar in terms of structure, regulations and staffing.”. In terms of specific antibiotic prescribing patterns, we have also added a sentence in the third paragraph in the section “Antibiotic use for respiratory tract infections” to highlight that our findings are similar to previous studies in Shandong province: “The high rate of parenteral antibiotics use, third-generation cephalosporins and fluoroquinolones in our study are concerning, but are consistent with the results of province-wide antibiotic consumption data that were recently reported in Shandong province using procurement records,[30] as well as previous studies at healthcare institutions in rural Shandong province [8].”

Comment: The prescriber-level analysis was limited to doctors with more than 50 upper respiratory infection prescriptions. This could have biased the results obtained. Explain.

Response: Three of the twenty-three doctors working at the village clinic had very small numbers of prescriptions with acute upper respiratory tract infection diagnoses (3, 12 and 19), which was in keeping with their very small numbers of overall prescriptions during the study period (5, 91 and 101). The explanation for two doctors (616 and 720) is that they retired towards the beginning of the study, and for doctor 618 the explanation is that he newly started working at the specific village clinic towards the end of the study. It is possible that their small number of AURI prescriptions may not be sufficiently representative of their actual prescribing practices for AURI, so we set a minimum of fifty AURI prescriptions during the 2.5 year study period for doctors to be included in presentations of overall ranges observed for antibiotic prescribing rates. We have expanded in the methods section “Data management and analyses”: “Prescriber-level analyses were limited to the 20/23 doctors with ≥ 50 AURI prescriptions during the study period, to ensure that the antibiotic prescribing rates calculated for each individual doctor were sufficiently representative of their practice. Of the three doctors with fewer than 50 AURI prescriptions, two retired from clinical practice near the beginning of the study period, and one newly started working at the specific village clinic at the end of the study period.”

In addition, we have further strengthened this prescriber-level analysis through using the Kruskal-Wallis H-test to assess whether the differences between prescribers were statistically significant, followed by pairwise Mann-Whitney U-tests post-hoc to assess how many of the pairwise comparisons between doctors’ antibiotic prescribing rates were statistically significant. We feel that for such analyses it is particularly important to only include prescribers with ≥ 50 prescriptions. We describe these additions in the methods section and results “The Kruskal-Wallis H-test showed that the variations in antibiotic prescribing rates for AURI prescriptions were statistically significant among

the 18 prescribers ($p < 0.001$). Pairwise Mann–Whitney U-tests were used as post-hoc tests to assess for differences in all possible pairwise comparisons between prescribers. This involved 153 pairs of prescribers, and the difference was statistically significant in 93 of these.”

Comment: There is no discussion about guidelines used by Chinese doctors. Some of the results found are not different from other countries and the wide variation across professionals is a characteristic observed in many other countries. However, the high rate of parenteral antibiotics, third-generation cephalosporins and fluoroquinolones found in this study merits a discussion.

Response: Thank you for suggesting that we raise these points. We have now included a paragraph in the section “Antibiotic use for respiratory tract infections”: “Clinical treatment guidelines are not in widespread use in village clinics in China [28], however there are essential medicine lists and lists of antibiotics which require higher authorisation levels to be prescribed. All but one antibiotic prescribed for AURIs were either included on the Chinese national essential medicines list [29], or on the list of supplemental essential medicines available in Shandong province; cefixime did not feature on either of these two lists and accounted for 1.6% of all antibiotics prescribed for AURIs during the study period. The high rate of parenteral antibiotics use, third-generation cephalosporins and fluoroquinolones in our study are concerning, but are consistent with the results of province-wide antibiotic consumption data that were recently reported in Shandong province using procurement records,[30] as well as previous studies at healthcare institutions in rural Shandong province.[8] In a survey we previously conducted at the village clinics included in this study, 74% of doctors reported thinking that parenteral antibiotics are more effective than oral antibiotics.[31]”

Comment: You refer pharyngitis and laryngopharyngitis as possible bacterial infections (Table 4). Is that correct?

Response: We were not clear enough in our choice of phrasing. These infections are most likely to be caused by viruses, but, compared with other upper respiratory tract infections, they have a higher likelihood of being caused by bacteria (and/or needing antibiotics). We have clarified this in the discussion in the last paragraph in section “Variations in antibiotic prescribing between prescribers”: “We did find that high antibiotic AURI prescribers were more likely than low AURI prescribers to prescribe antibiotics for the group of URTI diagnoses that are most likely to have bacterial causes (but where the most common causes are still viral). One possible explanation is that these prescribers could be more risk averse, and so are more likely than other doctors to prescribe antibiotics in cases where the cause could be bacterial.”

Comment: You highlighted the high consumption of antibiotics for gastrointestinal and dental infections compared to studies carried out in high-income countries but no references are provided. Not many papers on dental infections have been published in western countries.

Response: We have re-phrased and expanded this sentence in the discussion (section “Consultation patterns and overall antibiotic use”) to make our points clearer, and included references: “We found that urogenital infections and skin and soft tissue infections each accounted for less than 1% of all antibiotic prescriptions at the village clinics, whereas these conditions are typically the second and third most common reasons for antibiotic prescriptions in studies in primary care clinics from high income countries [17]. In contrast, dental conditions were the third largest group of diagnoses for which antibiotics were prescribed in our study. This may reflect the extreme paucity of dentists in rural China [11], and is potentially an important and under-reported source of antibiotic overuse.”

Comment: Did you analyze doctors’ characteristics when you compared the high and the low prescribers? For instance, use of rapid tests, time per consultation, etc.

Response: No, we did not collect such data that would have enabled this type of analysis. We fully agree it would be interesting to do so, particularly among a larger cohort of doctors.

Reviewer: 2

Reviewer Name: David Hyun

Institution and Country: Antibiotic Resistance Project, The Pew Charitable Trusts, United States

Please state any competing interests or state 'None declared': None declared

Comment 1. I recommend moving up the description included at the end of the manuscript in Methodological Considerations, that all clinic visits ends with a prescription, to the materials and methods section. As a reader not familiar with Chinese village clinics and their practices, I found myself questioning the authors' implied equivalence between the total number of prescriptions and the total number clinic visits for the denominator of the rates measured, until I found the clarification at the end of the manuscript.

Response: Thank you for this suggestion, we completely agree. We have added a sentence in the last paragraph in the background: "Village clinics are the first place that most rural residents seek healthcare, and almost all consultations result in patients receiving a prescription, in part because prescriptions are a way in which reimbursements for healthcare expenditures are calculated and administered.". We have also included a sentence in the abstract section "Design, setting and participants": "Almost all outpatient encounters at village clinics result in a prescription being issued."

Comment 2. A few clarifications will be helpful in the methods employed on translating the diagnoses captured from the e-prescription databases to ICD10 codes, given my unfamiliarity of how diagnoses are documented in the e-prescriptions from village clinics and since the crossmapping to ICD10 codes was conducted by just one researcher.

a. What were the qualifications of the independent researcher who conducted this coding translation? I'm assuming it was someone with clinical training or background?

Response: The independent researcher is a clinical doctor from China with additional training in public health (in Europe).

Comment 2. b. Was there a pre-defined list/mapping of Chinese symptoms/diagnoses to ICD10 that the independent researcher used systematically?

Response: We have clarified in the methods "All diagnoses on prescriptions were coded where possible according to the International Classification of Diseases, 10th Revision (ICD-10) by an independent researcher from China with a background in clinical medicine and public health: first, a list of unique diagnoses included on the prescriptions was created; second, these were matched with the standard ICD-10 Chinese translation provided by the National Health Commission of China that is used in hospitals; finally, these diagnoses were translated into English. In cases of uncertainty a shortlist of potential diagnoses was discussed between the first author (a clinical doctor and public health researcher from Europe) and the independent researcher until either agreement was reached or no ICD-10 code was assigned (e.g. when a symptom was listed instead of a diagnosis, or when it was unclear whether a diagnosis was acute or chronic); in total, these cases of uncertainty accounted for <10% of all prescriptions, and did not include any prescriptions that were ultimately classified as likely viral acute upper respiratory tract infections."

Comment 2.c. Does the e-prescription allow only one diagnoses entry from each visit? Other studies from other countries have shown that chart and billing documentation from outpatient visits, especially for acute respiratory conditions, frequently results in more than one diagnoses. And many of those multiple diagnoses have differing appropriateness for antibiotic prescribing (for instance URI and pneumonia diagnosis documented/coded in the same visit). Are there any limitations in the e-

prescription diagnosis data that could suggest the possibility that certain clinic visits with multiple diagnoses may have included a diagnosis that was antibiotic appropriate that was not captured in the e-prescription system (such as if the e-prescription system auto-populates the first diagnosis from a list of diagnoses in the electronic medical chart)?

Response: Thank you for this point. The e-prescription system only allows one diagnosis per prescription. If a patient attended with more than one infection, then our understanding is that the doctor would usually write the more severe infection as the diagnosis. We therefore think it is very unlikely that a significant number of the antibiotics prescribed for diagnoses of acute upper respiratory tract infections and common colds (the main focus of this paper) were actually prescribed for other infections. In contrast, we have found that antibiotics were included on some prescriptions with non-infection-related diagnoses, such as hypertension. We have purposefully avoided classifying these as “inappropriate antibiotic use” for this reason. We have now added a couple of sentences to the section “methodological considerations”:

“Finally, only one diagnosis can be included per prescription on the electronic prescribing system. It is possible that some antibiotics prescribed for likely viral acute upper respiratory tract infections are actually treatment for other infections that patients presented with at the same time; however, doctors would usually choose to write the more severe diagnosis, so we believe this is likely only relevant to a very small number of cases and will not have significantly affected our main findings.”

Comment 2. d. Would like to know about the “coding error” that led to the exclusion of prescriber from the analysis – was it a coding error originated from the e-prescription system or was it a coding error by the independent researcher during the ICD10 code crossmapping process?

Response: This error was created at the level of the village clinic. In brief, one of the two doctors working at this clinic was not very proficient with computers and so he usually wrote his prescriptions by hand. His colleague doctor then entered these written prescriptions on the electronic system, but the colleague did not always use the correct prescriber’s account (which has no administrative consequences because reimbursements for work are conducted at a village clinic level). Once we detected this error we checked to make sure this was not the case at other clinics. We have revised the sentence in the methods section “Data management and analyses” to explain the error more clearly: “We identified an error in the coding of prescriptions in village clinic 1 which was due to one doctor (101) being responsible for entering the majority of prescriptions onto the electronic system using their account, irrespective of who the actual prescriber was, because of the low computer literacy of the other doctor (102). We therefore could not be confident of which doctor was responsible for which prescription, so all 433 prescriptions from the two doctors working at this clinic were excluded from prescriber-level analyses. This error was not present at other village clinics.”

Comment 3. Reviewing the N values of the individual prescribers in Appendix Table 2, it would be helpful to have the following questions addressed in the text – mostly relating to additional information on the individual provider practice characteristics.

- a. Looking at the two excluded prescribers (616 and 618) due to their very low N values for AURIs, it raises the question on why their Ns were so low to begin with especially considering they were from the same clinic and the only other prescriber in the same clinic (617) was responsible for over 90% of the AURI visits. Were the two excluded prescribers part-time providers? Are they seasonal providers (doesn’t see patients during the winter?) Are there any other demographics/practice factors that could account for this significance difference between providers in patient population within the same clinic?
- b. Are any of these factors applicable to other village clinics where it could have influenced patient population differences between the providers from the same clinic?

Response: These two prescribers did have low numbers of AURI prescriptions (12 and 19), but this was in keeping with their very small numbers of overall prescriptions throughout the study period (91 and 101). The explanation is that doctor 617 was the main doctor at this village clinic throughout the study period. Doctor 616 retired towards the beginning of the study and was only working part-time beforehand, so they contributed a small number of total prescriptions. Doctor 618 started working at the village clinic towards the end of the study period, and so again only contributed a small number of total prescriptions. We also know that doctor 720 (in a different village clinic) retired early on during the study, and so also contributed a very small number of total prescriptions. We have written in the methods section "Data management and analyses": "Prescriber-level analyses were limited to the 20/23 doctors with ≥ 50 AURI prescriptions during the study period, to ensure that the antibiotic prescribing rates calculated for each individual doctor were sufficiently representative of their practice. Of the three doctors with fewer than 50 AURI prescriptions, two retired from clinical practice near the beginning of the study period, and one newly started working at the specific village clinic at the end of the study period."

We have no reason to believe that there are other factors that account for significant differences between providers in terms of patient population within the same clinic.

Comment 4. A couple of suggested bibliographies to add in the discussion around individual provider variation in antibiotic prescribing:

Stenehjem E, Wallin A, Fleming-Dutra KE, Buckel WR, Stanfield V, Brunisholz KD, Sorensen J, Samore MH, Srivastava R, Hicks LA, Hersh AL. Antibiotic Prescribing Variability in a Large Urgent Care Network: A New Target for Outpatient Stewardship. Clin Infect Dis. 2019 Oct 23. pii: ciz910. doi: 10.1093/cid/ciz910. [Epub ahead of print]

Jung S, Sexton ME, Owens S, Spell N, Fridkin S. Variability of Antibiotic Prescribing in a Large Healthcare Network Despite Adjusting for Patient-Mix: Reconsidering Targets for Improved Prescribing. Open Forum Infect Dis. 2019 Jan 18;6(2):ofz018. doi: 10.1093/ofid/ofz018.

Response: Thank you for suggesting these papers, we have now referenced them in the discussion on variations in antibiotic prescribing between prescribers: "Jung et al found an interquartile range of 27% to 60% in antibiotic prescribing rates for RTIs among 109 prescribers in a large healthcare network in the US;[39] a study conducted in a separate US healthcare network of urgent care clinics found an interquartile range in antibiotic prescribing rates among physicians of 7% to 28% for respiratory encounters where antibiotics are not indicated.[40]"

VERSION 2 – REVIEW

REVIEWER	Carl Llor University Institute in Primary Care Research Jordi Gol, Via Roma Health Centre, Barcelona. I report having received research grants from Abbott Diagnostics.
REVIEW RETURNED	10-Mar-2020
GENERAL COMMENTS	All the queries have been satisfactorily responded. The paper reads now much better.

REVIEWER	David Hyun The Pew Charitable Trusts, United States
REVIEW RETURNED	15-Mar-2020

GENERAL COMMENTS	The responses provided for the questions raised from the initial review have been very helpful, and they have been appropriately incorporated into the manuscript text edits that provide more clarity in the methodology of the study as well as describing potential limitations in interpreting the results.
---